

**Intraseasonal Variability and Eddy-Induced Structural Modulation of the North**
**Pacific Intermediate Water Revealed by Multi-Mooring Observations**

Qiang Ren[1,2], Yansong Liu[1,2], Shumin Tu[3], Wei Huang[3], Feng Nan[1,2,4], Ran Wang[1,2],
Xinyuan Diao[1,2,4], Jianfeng Wang[1,2], Xinchuang Liu[1,2], Zifei Chen[1,2]*,Fei Yu[1,2,4]*
[1] Key Laboratory of Ocean Observation and Forecasting, Institute of Oceanology,
Chinese Academy of Sciences, Qingdao, China
[2] Key Laboratory of Ocean Circulation and Waves Institute of Oceanology, Chinese
Academy of Sciences, Qingdao, China
[3] Laboratory of Low Frequency Electromagnetic Communication Technology with the
WMCRI, CSSC, Wuhan, China
[4] College of Marine Sciences, University of Chinese Academy of Sciences, Qingdao
266071, China

**Corresponding authors:** Fei Yu (yuf@qdio.ac.cn) and Zifei Chen (chenzifei@qdio.ac.cn)



**Abstract:**

The North Pacific Intermediate Water (NPIW) plays a crucial role in modulating oceanic thermohaline circulation and biogeochemical processes. However, limited continuous observations have hindered the understanding of its short-term variability and structural response to mesoscale processes. This study investigates the intraseasonal structural variability of the NPIW and its modulation by mesoscale eddies, based on long-term mooring observations from three sites (M1–M3) across the western Pacific. The thickness of the NPIW displays substantial intraseasonal variability, dominated by an approximately 80-day period that is coherent among all mooring sites. Unlike previous studies that mainly focused on temperature and salinity anomalies, this work introduces NPIW thickness as a new structural diagnostic parameter to capture the vertical compression and expansion of the intermediate layer induced by eddy activity. The analysis identifies a strong inverse correlation between layer thickness and isopycnal-averaged salinity, demonstrating that anticyclonic (cyclonic) eddies correspond to thinner (thicker) and more saline (fresher) intermediate layers. Spatial composites further reveal that thickness variability is most pronounced near the western boundary, which may be associated with locally complex water mass exchange and mixing driven by eddies. These findings provide the first quantitative evidence of intraseasonal variability in NPIW thickness and highlight its role as a key indicator for diagnosing mesoscale–intermediate layer interactions in the North Pacific.

**Index Terms and Keywords**
Mesoscale eddies drive intraseasonal variability of NPIW
Thickness and salinity reveals structural responses of NPIW
Eddy-induced mixing reshapes NPIW properties along the western boundary.

# 1 Introduction

The North Pacific Intermediate Water (NPIW) is a pivotal component of the North Pacific's water mass and extensively studied due to its significant role in climate dynamics and oceanic processes (Talley, 1993; Masuda et al., 2003; You et al., 2003; Gong et al., 2019; Nishioka et al., 2020). This water mass originates in the northwestern subtropical gyre, within the transition zone between the Kuroshio Extension and the Oyashio front, is characterized by its low salinity and relatively cooler temperatures at depths of approximately 400 to 1200 meters, also its density is centered around 26.8 $\sigma_\theta$





isopycnal, with a salinity minimum about 34.1 to 34.3. (Talley, 1993, 1995; Yasuda et
al., 1997; You et al., 2003; Masujima et al., 2009). NPIW is an important intermediate
water mass connecting the upper and deeper layers of the ocean, and has important
implications for physical, biological, chemical, and ecological processes such as
dissolved oxygen, nutrient distribution, and thermohaline transport (Talley et al., 1993;
Hansell et al., 2002; Auad et al., 2003; Tsunogai et al., 2002; Ohkushi et al., 2003; Zhou
et al., 2022). NPIW also plays an important role in global biogeochemical fluxes such
as carbon and nutrient cycling (Tsunogai et al., 2002; Ohkushi et al., 2003).
The distribution and transport pathways of NPIW have been a focal point of
oceanographic research, many studies have shown that the NPIW is widely distributed
in the North Pacific Ocean, and that it is transported by complex water masses and
circulation (Qiu, 1995; Ueno & Yasuda, 2004; Yasuda, 2004; Gordon and Fine, 1996;
Kashino et al., 1996; Kashino et al., 1999; Yuan et al., 2022). You (2003) found that
NPIW originates from the subpolar regions of the North Pacific and propagates through
the eastern subtropical gyre towards the Indonesian Through flow. As a result, NPIW
can be found in eastern Japan, eastern Taiwan, the West Philippine Basin, and the
intermediate region of the North Pacific Ocean (You. 2003; Fujii et al., 2013). Based
on previous studies, we determined the approximate distribution of the NPIW from the
WOA13 showed in Fig. 1, and this figure is similar to the results of You (2003), we
reproduce the depth distribution characteristics of NPIW in different regions. The
NPIW was found to have different depths in different regions, with shallower depths in
the western boundary region and deeper depths in the middle of the North Pacific Ocean.
Since NPIW is one of the most important water masses in the global ocean, most of
studies focus on its seasonal, interannual or interdecadal variations in different regions,
and these variability is largely influenced by multi-scale ocean-atmosphere interactions
(Masuda et al., 2003; Ohshima et al., 2010; Bingham& Lukas., 1995; Solomon et al.,
2003; Qiu et al., 2011; Van et al., 1993; Sugimoto et al., 2022; Li et al., 2023). However,
the majority of the studies mentioned above focus on time scales exceeding a few
hundred days, and also the NPIW are located in the deep layers below the subsurface,
where direct and long-term observations are difficult. More than that, there is often a
large bias in the salinity representation of the water masses in the intermediate layer of
the model data, there are very few studies of intraseasonal variations in the NPIW.
Mesoscale eddies are widely found in the oceans, with lifetime ranging from a few days
to several hundreds of days, and radii of up to several hundreds of kilometers in the mid
latitude (Wyrtki et al., 1976; Richardson, 1983; Robinson, 1985; Chelton et al., 2007;
Chelton et al., 2011; Zhang et al., 2014; Wunsch et al., 2007; Martínez-Moreno et al.,
2021). A large number of observational studies have shown that eddies can affect depths
of up to kilometers, that there are significant differences in the three-dimensional
structural features within anticyclonic and cyclonic eddies, and that mesoscale eddies
produce different temperature and salinity anomalies by causing uplift or subsidence of
the isopycnals. (Zhang et al., 2015; Thoppil et al., 2011; Zhang et al., 2016; Zhang et
al., 2015; George et al., 2021; Waite et al., 2016; Hausmann et al., 2017). Within the
range of NPIW generation, propagation and distribution, there is also a high incidence
of mesoscale eddies, it is therefore of great interest to investigate whether mesoscale



eddies have an impact on the NPIW in different regions and with different thermohaline characteristics. In a localized area along the western boundary, Mensah et al. (2015) examines the intraseasonal to seasonal variability of intermediate water east of Luzon and Taiwan by hydrographic data from several cruises, it deduced a possible relationship between the eddies and the intermediate water from SLA data. Also, Wang et al. (2016) revealed that the semiannual variability of water masses at the northern and southern hemispheric convergence near 8° N related to mesoscale eddies. Next, Ren et al. (2022) found an intraseasonal variability of the IW of ~80 days from direct observations of the subsurface moorings east of Taiwan, and that this variability is associated with mesoscale eddies. These studies reveal the complex variability of NPIW in the western boundary region, which may be extensively influenced by local water masses such as the South China Sea Intermediate Water Mass and the Kuroshio Intermediate water mass. Also these studies can illustrate some of the effects of eddies on IW, but they are insufficient to demonstrate the widespread and persistent existence of NPIW's intraseasonal variability characteristics, which is one of the most important links between high-frequency variability and climate-scale cycles of change.

Previous studies have significantly advanced the understanding of the formation, distribution, and variability of the North Pacific Intermediate Water (NPIW). However, most of these studies have primarily focused on temperature and salinity anomalies, offering limited insight into the structural response of NPIW to dynamic processes such as mesoscale eddies. As noted by Nakanowatari et al. (2015), model-based analyses often exhibit large uncertainties due to the scarcity of long-term in situ data, constraining their ability to accurately represent the vertical structure and temporal evolution of NPIW. These limitations highlight the necessity of direct, continuous mooring observations to resolve intraseasonal processes that can strongly influence the intermediate-layer structure and mixing. To address these gaps, the present study introduces intermediate water thickness as a structural diagnostic parameter to characterize the physical adjustment of NPIW under mesoscale eddy forcing. Thickness, defined as the vertical extent of the low-salinity core, serves as an integrated indicator of baroclinic adjustment, isopycnal displacement, and mixing intensity. Moreover, variations in NPIW thickness can directly influence the vertical distribution of nutrients and dissolved oxygen, linking physical dynamics with mid-depth biogeochemical and ecological processes. Changes in the structural extent of NPIW may also modulate carbon storage and ventilation pathways, highlighting the broader climatic and ecological implications of mesoscale-driven structural variability in the intermediate ocean.

By combining long-term mooring observations from three distinct sites (M1–M3) across the western Pacific, this study provides the first quantitative evidence of intraseasonal variability in NPIW thickness and its strong inverse relationship with salinity. This structural perspective complements traditional thermohaline analyses and enables a more comprehensive understanding of how mesoscale eddies reshape the hydrographic properties and vertical structure of the NPIW.



## 2 Data and Methods

### 2.1 Mooring data

To investigate NPIW variability, three mooring systems were deployed in the northwestern Pacific (Fig. 1). The locations, observation periods, and equipment setups of these three moorings, M1, M2, and M3, are described as follows. The mooring M1 is located at 146°E and 25°N, with an observation period from April 2017 to June 2018; M2 is located to the east of Taiwan on the western boundary, at 122.67°E and 22.3°N, with an observation period from August 2019 to December 2020; while M3 is located at 126°E and 18°N, with an observation period from January 2016 to June 2017. Each mooring was instrumented with conductivity–temperature–depth sensors (Sea-Bird Electronics SBE 37) installed at 100 m vertical spacing between depths of 400 and 1000 m, with all instruments programmed to record data at 10-minute intervals. The deployment depths were carefully designed to span the upper and lower boundaries of the NPIW, ensuring adequate vertical representation of its structure. While slight vertical motion of the mooring line may occur under strong currents, the CTDs moved synchronously with the line, and all observations were converted into fixed pressure-based depths using pressure sensor data. This adjustment effectively minimizes any potential depth uncertainty due to mooring motion. Local linear interpolation between adjacent CTD sensors bracketing the 34.3 psu value was applied to estimate the depths of the isohalines for thickness calculation. Interpolation applied for visualization purposes is separate and does not affect the quantitative analyses. All figures presented in this paper display the interpolated fixed-depth data, and also processed for daily averages after deleted the abnormal value.

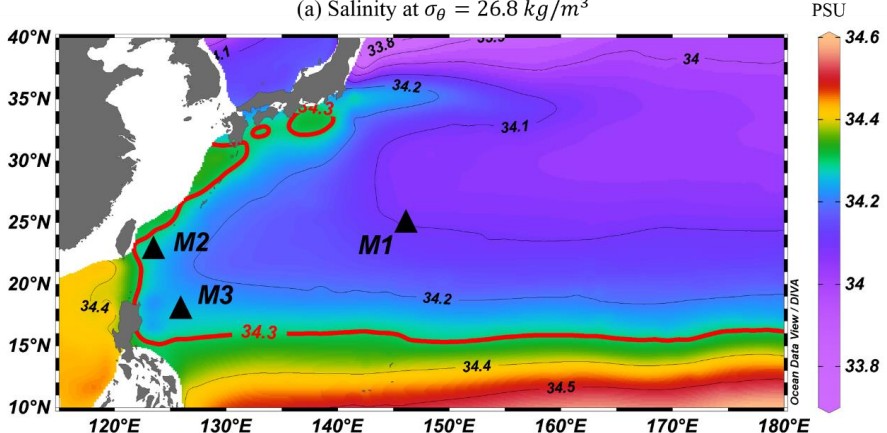

(a) Salinity at $\sigma_\theta = 26.8\ kg/m^3$



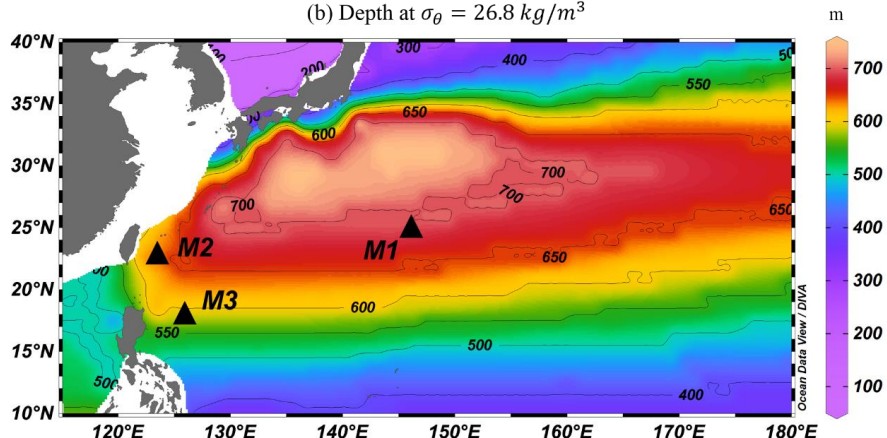

Figure 1. Distribution of salinity (a) and depth (b) along the $26.8\sigma_\theta$ isopycnals at Northwestern Pacific. The color shading and black line in (a) and (b) represent the salinity and depth, respectively. The red contours of 34.3 psu is represent the NPIW range from Talley et al. (1993) and You et al. (2003). The black triangle is the mooring location: Mooring 1 (M1), Mooring 2 (M2) and Mooring 3 (M3). Salinity and depth in the Fig. 1 are taken from climatological averaged data from World Ocean Atlas 23, and plot with *Ocean Data View.*

## 2.2 World Ocean Atlas 2023

The spatial distribution of NPIW in the northwestern Pacific was examined using the World Ocean Atlas 2023 (WOA23). Produced by NOAA's National Oceanographic Data Center–Ocean Climate Laboratory (The data available online at: https://www.ncei.noaa.gov/products/world-ocean-atlas), this dataset consists of objectively analyzed climatological fields derived from in situ observations, including temperature, salinity, dissolved oxygen, and inorganic nutrients at 102 standard depth levels in the global ocean (Reagan et al., 2024).

## 2.3 The Copernicus Marine Environment Monitoring Service (CMEMS) data.

In this study, two Copernicus Marine Environment Monitoring Service (CMEMS) products were utilized.

(1) Sea Level Anomaly (SLA) and Geostrophic Currents data

We used the Global Ocean Gridded L4 Sea Surface Heights and Derived Variables Reprocessed Dataset (SEALEVEL_GLO_PHY_CLIMATE_L4_MY_008_057, https://doi.org/10.48670/moi-00145), provided by CMEMS (https://marine.copernicus.eu/). This altimetry product merges multi-mission satellite observations and provides global gridded fields of sea level anomaly (SLA), absolute dynamic topography, and geostrophic currents. The data have a spatial resolution of 1/4° and daily temporal resolution, covering the observation periods of all three subsurface moorings.



(2) Temperature and Salinity Reanalysis Data
To analyze subsurface temperature and salinity variability around the mooring sites, we
employed the Global Ocean Physics Reanalysis Product
(MULTIOBS_GLO_PHY_TSUV_3D_MYNRT_015_012,https://doi.org/10.48670/m
oi-00052), a Level-4 global reanalysis distributed by CMEMS. This product provides
three-dimensional fields of temperature, salinity, potential density, and geostrophic
currents on a regular 1/8° grid, spanning from the surface to 5500 m with 50 vertical
levels. The product is generated by combining in situ and satellite observations on a
global scale. The available record covers the period from January 1993 to the present,
with temporal resolutions of weekly and monthly (Guinehut et al., 2012; Mulet et al.,
236 2012).

## 3 Result

### 3.1 Hydrographic and temporal characteristics of NPIW

To gain an initial understanding of the NPIW characteristics at the locations of our three
deployed moorings, we employed the WOA database to create decadal average maps
of salinity and depth distribution on the 26.8 $\sigma_\theta$ isopycnal, as illustrated in Fig. 1. The
NPIW shows significant local variability, with the NPIW showing lower salinity values
and its core depth of approximately 34.1 psu and 700 m near M1 mooring site,
respectively. NPIW in the western and southern parts of the distribution, the minimum
salinity of the NPIW increases and its depth is relatively shallow. Near the M2 mooring
location, the low salinity value and depth adjust to about 34.25 psu and 600 meters,
respectively. Based on the NPIW range determined by the 34.3 psu contour of the
salinity definition, M3 near 18°N, which can be seen in Fig. 1 to be located close to the
south edge of the NPIW distribution, has a salinity minimum value close to that at M2,
but the depth of the low salinity core becomes further shallower to ~550 meters. Thus,
the moorings utilized in this study have effectively observed NPIW, capturing its
significant spatial and temporal variability across different regions.
Observations from the M1 mooring over more than a year, as shown in Fig. 2a, reveal
that the low salinity core of the NPIW has an average depth of approximately 700
meters, fluctuating within the range of 26.4 to $27\sigma_\theta$ isopycnal, with the minimum
salinity value being around 34.15 psu. The observed salinity minima at M1 were also
found to be slightly higher compared to the climatological averaged data showed in Fig.
1a. Additionally, significant temporal variations in salinity were observed at depths of
400-900 meters by the M1 mooring. The M2 mooring located east of Taiwan near the
western boundary, observed salinity below 400 meters as depicted in Fig. 2c. The low
salinity core varied between the 26.6-26.8 $\sigma_\theta$ isopycnals showing more significant
changes than those observed at M1, with average minimum salinity values and depths
of approximately 34.2 psu and 600 meters, respectively, which are higher than the
minimum salinity values observed at the M1 location. And it is interesting to note that
the low salinity core of M2 is apart, such as the salinity measured in April-May 2020 in
Fig. 2c, which is close to 33.6 psu, splitting the low salinity core with a salinity value





of around 34.2 psu. At the more southerly M3 mooring, as illustrated in Fig. 2e, the low
salinity core also apart with seven significant low salinity events observed over a year.
The average minimum salinity value between the 26.6-26.8 $\sigma_\theta$ isopycnals was 34.3
psu, with corresponding temperatures and depths of approximately 8°C and 550 meters,
respectively. A distinctive feature of M3 was that the depth of the NPIW's low salinity
core was shallower than that at M1 and M2, and the minimum salinity was significantly
higher than M1 and M2. The results of NPIW observed by the three differently
positioned subsurface mooring are basically consistent with the spatial distribution
characteristics of NPIW in the North Pacific Ocean in the WOA data.
Upon comparing Fig. 2a, 2c, and 2e, it appears that the intermediate water masses at
the M2 location exhibit greater variability, while those at the M1 location show
relatively weaker variations. From the corresponding salinity standard deviation plots
(Fig. 2b, 2d, and 2f), it is observed that the M1 mooring displays the smallest standard
deviation at the NPIW core depth of approximately 700 meters, indicating higher
stability in intermediate layer salinity. Conversely, the salinity at the levels of NPIW for
M2 and M3 shows greater variability. The largest standard deviation in salinity at the
mooring M2 is 0.7 psu at around 600 meters, shown in Fig. 2d, while a significant
standard deviation in salinity around 0.3 psu is observed between 500-600 meters from
mooring M3. This variability in salinity in intermediate layer is also depicted in the T-
S (Temperature-Salinity) plot in Fig. 3, where the range of salinity changes at the
mooring M2 is the largest among the three observed locations, ranging from 34.13 psu
to 34.35 psu, with M1 showing the smallest variation. Differences in standard
deviations also illustrate the variability of NPIW changes across regional locations,
with the least variability at 25°N, possibly related to its deeper depth. The relatively
strong intermediate-layer salinity variability near M2 is likely associated with complex
local circulation, where intermittent influences of South China Sea Intermediate Water
(SCSIW), a salinity-minimum water mass (~34.4 psu at ~500 m) are modulated by the
Kuroshio and mesoscale eddies (Menash et al., 2015; Ren et al., 2022). Overall, the
measurements from the subsurface moorings show a change in the minimum salinity in
the mid-ocean region, which also corresponds to a salinity within the range defined by
the NPIW, and this change also reflects the fluctuations in low salinity core of the NPIW.

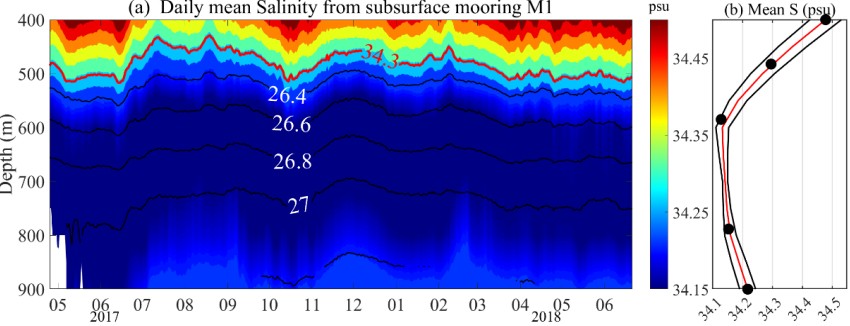


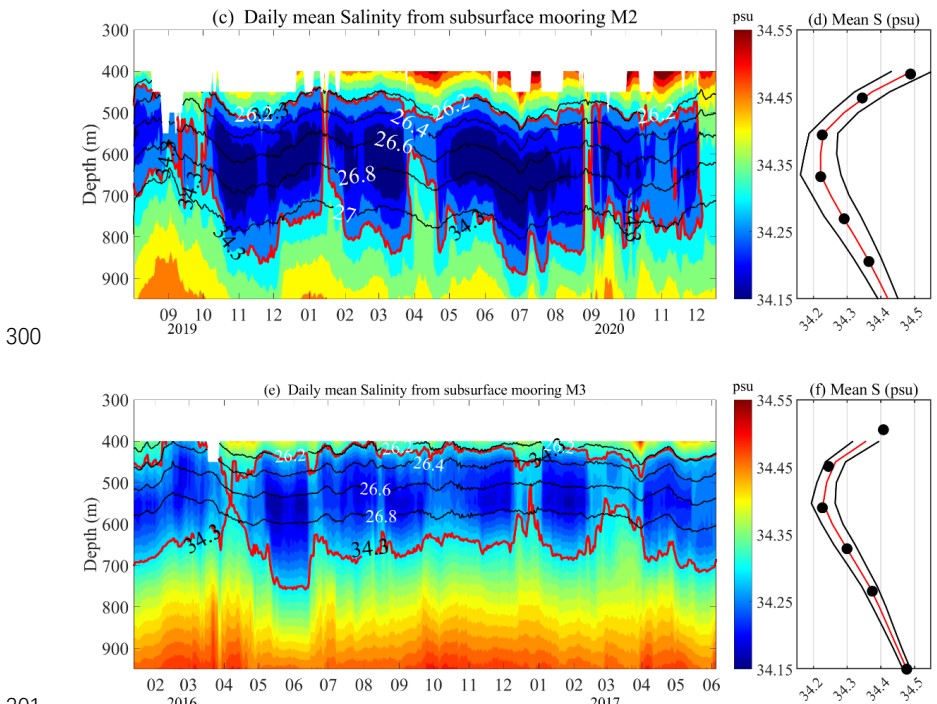

Figure 2. The observation of salinity from subsurface mooring. (a), (c) and (e) represent time series plots of salinity measured at different observation times for the three moorings, respectively. M1 is observed from April 2017-June 2018, M2 is observed from August 2019-December 2020 and M3 is observed from January 2016- June 2017. Color shading and the black lines represent the salinity and 26.2 to 27.0 $\sigma_\theta$ isopycnal, and also red line represent the 34.3 psu in (a), (c) and (e), respectively. In (b), (d) and (f), the red line and black line represent the mean salinity and standard deviation of salinity over the observation period, respectively. The black circle in (b), (d) and (f) are represents the average depth of deployed CTDs in subsurface moorings.

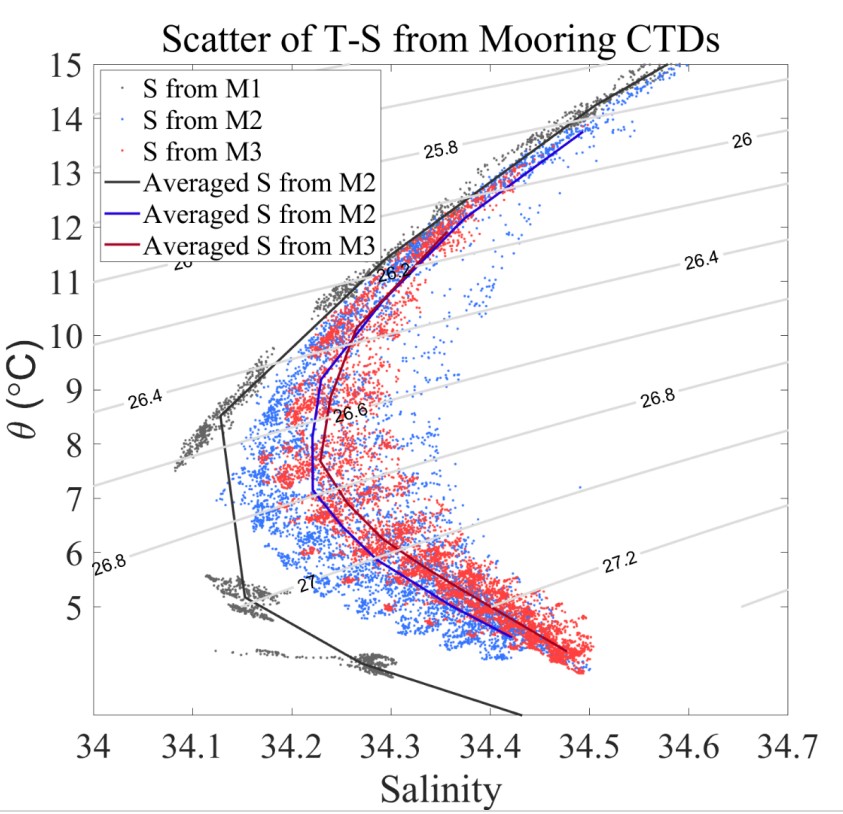

Figure 3. (a) The Temperature-Salinity plot from mooring CTDs. The blue, green and orange points are represent the M1, M2 and M3 measurements temperature and salinity, the black, green and red curves are represent the averaged M1, M2 and M3 measurements T-S, respectively. The blue dashed box is represent the range of NPIW defined from Talley et al. (1993) and You et al. (2003).

## 3.2 Intraseasonal Variations of NPIW Thickness and Salinity

Given that salinity alone cannot fully capture the structural variability of the North Pacific Intermediate Water (NPIW), this section examines the variations in both layer thickness and isopycnal-averaged salinity to better characterize its dynamic evolution. The mean salinity between the 26.4 and 26.9 σθ isopycnals was used to represent the NPIW core, a range widely recognized in previous studies as encompassing both the central low-salinity layer and its surrounding transition zones. The NPIW thickness was defined as the vertical distance between the depths where salinity equals 34.3 psu, a threshold corresponding to the upper limit of the NPIW core in the western North Pacific. Using the 34.3 psu isohaline as a tracer-based boundary provides a consistent, physically meaningful measure of the intermediate layer's volumetric extent, enabling comparative analysis of thickness variations across time and mooring sites. At mooring site M1, a threshold of 34.2 psu was used due to the presence of a fresher intermediate layer, consistent with regional hydrographic features.




Figure 4 presents the time series of NPIW thickness (red line) and average salinity
between isopycnal surfaces (blue line) at three mooring sites (M1, M2, and M3). At M1
(Fig. 4a), during 2017–2018, the NPIW thickness generally fluctuated between 250 and
400 m, with an average around 300 m. The corresponding average salinity remained
consistently below 34.2 psu, closely matching the classical NPIW characteristics. A
pronounced minimum in salinity was observed in October–November 2017, coinciding
with a local maximum in thickness exceeding 350 m. At M2 (Fig. 4b, 2019–2020), the
NPIW thickness exhibited more dramatic fluctuations, varying abruptly between 0 and
350 m. Notably, in October 2019, and February and September 2020, the thickness
dropped from around 300 m to nearly zero, accompanied by a sharp increase in salinity
exceeding 34.35 psu. This suggests a rapid erosion or replacement of intermediate water
properties, likely driven by more active eddy activity or local mixing processes. At M3
(Fig. 4c) showed generally thinner NPIW, mostly between 100 and 250 m, with an
average thickness of around 200 m. Similar to M2, the thickness at M3 was also marked
by strong short-term fluctuations. Although the absolute thickness and salinity values
varied across the three sites, all stations demonstrated consistent a clear and persistent
inverse relationship between thickness and salinity. Periods of increased NPIW
thickness were generally associated with decreased salinity. This structural variability
implies that intermediate water changes may be modulated by local mesoscale eddies
or mixing between water masses.

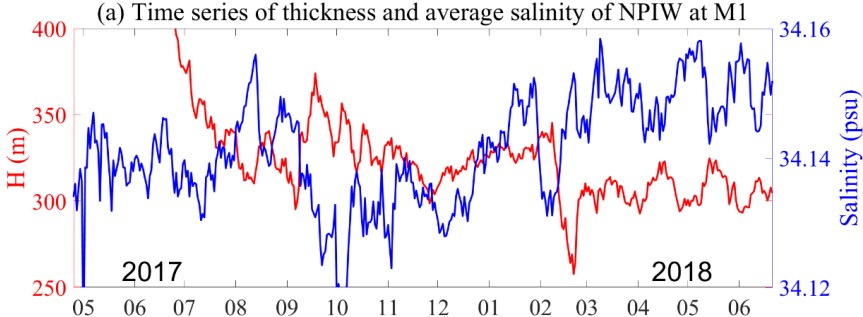


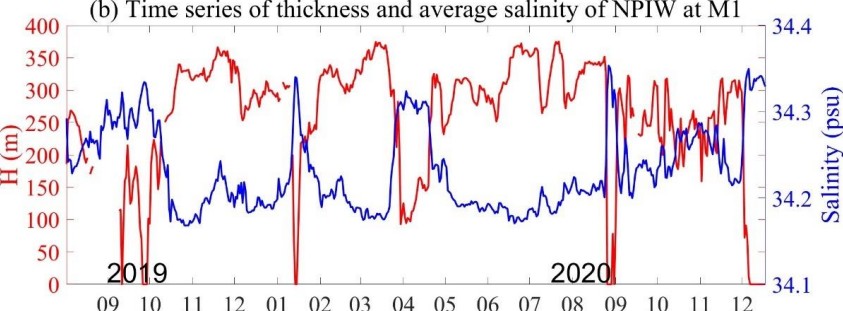


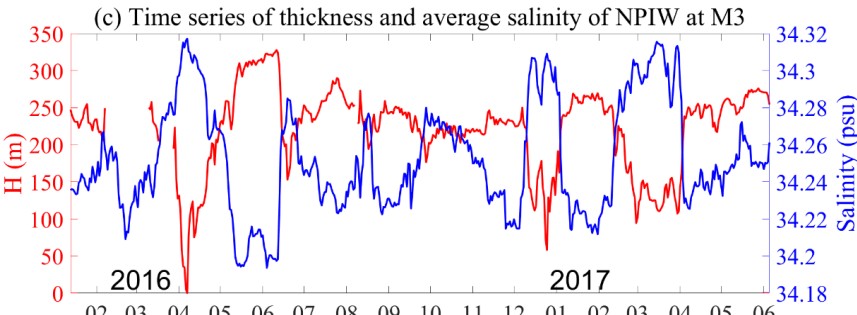

Figure 4. Time series of thickness and averaged salinity of NPIW at M1 (a), M2 (b) and
M3 (c). The NPIW thickness was defined as the vertical distance between the depths
where salinity equals 34.3 psu. The average salinity between the 26.4 and 26.9 σθ
isopycnal was used to represent the salinity characteristics of the NPIW.

To further investigate the temporal characteristics underlying the observed thickness
and salinity fluctuations, particularly their dominant time scales, a wavelet analysis was
performed on both salinity and thickness at each mooring site. As described in the
previous section, the relationship between NPIW thickness and isopycnal-averaged
salinity effectively captured the internal structural response of the intermediate layer.
However, when examining temporal variability and dominant periodicities, a fixed-
depth averaged salinity was used instead of isopycnal averaging. This approach allows
the salinity signal to incorporate the vertical displacement of isopycnals induced by
mesoscale eddies, thereby enabling a more direct comparison with the SLA time series
and identification of intraseasonal oscillations.
Although the wavelet spectra were also calculated for NPIW thickness, the results
exhibited nearly identical dominant periods and spectral power to those of salinity,
confirming the tight coupling between thickness and salinity variations. Therefore, only
the wavelet spectra of salinity are shown for brevity. The results at site M1 (Fig. 5a–b)
indicate a pronounced intraseasonal variability with a dominant period of
approximately 70–80 days, consistent with the eddy-related SLA fluctuations observed
in the same region. While this intraseasonal variation cycle exhibits temporal variability,
it was more significant from May 2017 to April 2018, while the signal strength of the
cycle significantly decreased after April 2018. Fig.5c-d represent the results of the
wavelet analysis of averaged salinity at 500 m to 800 m at M2, with a similar ~80 days
period as on the M1, the intraseasonal signals at M2 also exhibit variability during
different observation periods. During the observation period from September 2019 to
August 2020, the variability period appears to be longer about 80 days showing in Fig.
5c. The observation results from mooring M3 shown in Fig. 5e-f, indicate relatively
stable intraseasonal variation periods of 70-80 days throughout the observation period.
These results collectively demonstrate that the structure and properties of the NPIW
exhibit robust intraseasonal variability across different locations, reflecting a common




intraseasonal signal that may be linked to regional mesoscale dynamics or other oceanic
processes.

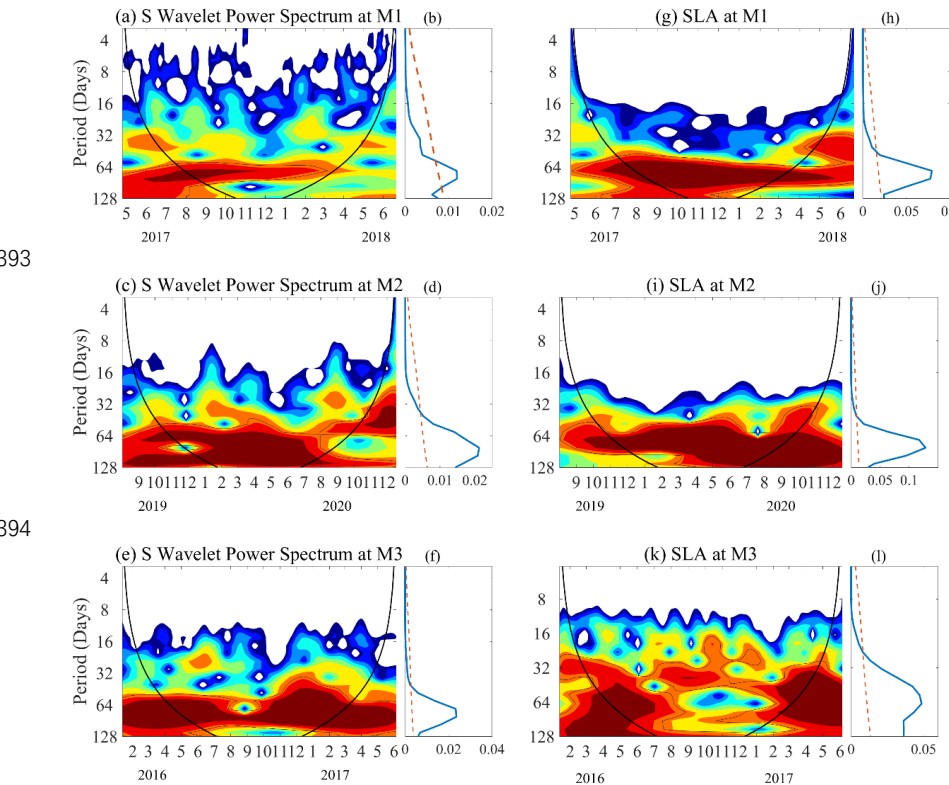

Figure 5. The wavelet power spectrum for salinity from 500 to 800 m at M1, from 500
to 800 m at M2 and from 500-700 m at M3 in (a), (c) and (e), respectively. (b), (d) and
(f) are the corresponding global spectrum of salinity in (a), (c) and (e), with the red
dashed line indicating the critical value at the 95% confidence level. (g), (i) and (k) are
the wavelet power spectrum for Sea Level Anomaly at the mooring site M1, M2 and
M3, also the (h), (j) and (l) are the corresponding global spectrum of SLA.

**3.3 Influence of SLA on the Salinity Structure of NPIW**
In the previous section, a strong inverse relationship between NPIW layer thickness and
isopycnal-averaged salinity was identified. This suggests that salinity can serve as an
effective proxy for structural changes in the intermediate water layer. Therefore, in this
section, we focus on the relationship between salinity and SLA, in order to indirectly
assess the influence of mesoscale eddies on the structural variability of the NPIW. In
the western North Pacific, mesoscale eddies are recognized as a major source of
intraseasonal signals (Zhou et al., 2021). To establish a possible link, we first examined
whether SLA, as a surface manifestation of mesoscale eddies, exhibits a dominant
intraseasonal period. As shown in Fig. 5g–l, wavelet analysis reveals a dominant 60–
80-day periodicity in SLA across all three mooring sites (M1–M3), which aligns closely




with the intraseasonal salinity variations previously identified. This temporal coherence
suggests a potential coupling between SLA and NPIW variability.
To further clarify their relationship, we applied a 20–120-day band-pass filter to the
salinity and temperature data in the intermediate layer and calculated correlation
coefficients with SLA (Fig. 6). At the M1 mooring location, SLA shows moderate
positive correlations of 0.55 with temperature and 0.45 with salinity, correlation
coefficients line within the 95% confidence bounds indicating that the estimated
correlations are statistically robust. The corresponding T–S diagram (Fig. 7a) supports
this pattern, indicating that relatively low temperature and salinity (or high temperature
and salinity) correspond to negative (or positive) SLA events. Given the strong salinity–
thickness relationship established earlier, these findings also imply that mesoscale
eddies may indirectly influence NPIW layer thickness through modulating
thermohaline properties. At M2, correlation coefficients between SLA and
temperature/salinity are slightly weaker (0.4 and 0.3, respectively; Fig. 6b), but the
positive pattern remains consistent, as confirmed by the T–S plot (Fig. 7b). At the M3
site, both temperature and salinity show a moderate positive correlation with the
corresponding SLA, where the correlations were found to be 0.47 and 0.45, respectively,
indicating that variations in the thermal–haline structure at this location are likewise
modulated by sea level anomalies. For example, periods of strongly negative SLA (e.g.,
April–May 2017) coincide with relatively fresh and cold intermediate waters (salinity
down to 34.2 psu), while periods of positive SLA (e.g., April 2016 and May–June 2017)
are associated with warmer and saltier conditions (salinity up to 34.3 psu).
Although the correlations between temperature, salinity, and SLA at these stations are
only moderate, the figures clearly show that when SLA reaches relatively large values,
the variations in temperature and salinity tend to become more coherent. These findings
collectively indicate that SLA most likely reflecting the presence and evolution of
mesoscale eddies, also is linked to intraseasonal variations in intermediate water
properties across all mooring sites. Coupled with the observed salinity–thickness
relationship, this suggests that the influence of SLA likely extends beyond simple
thermohaline anomalies and plays an important role in shaping the structural variability
of the NPIW as well.

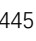

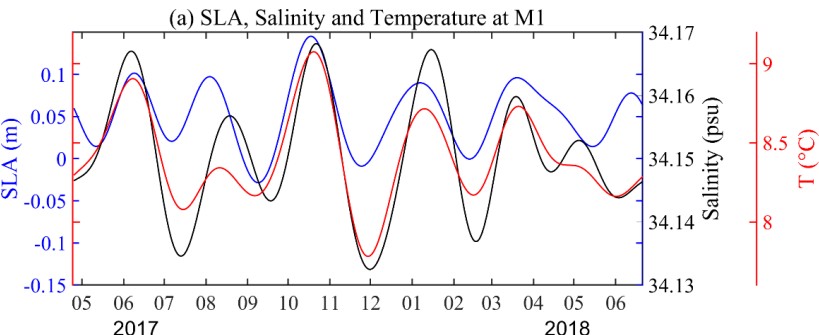






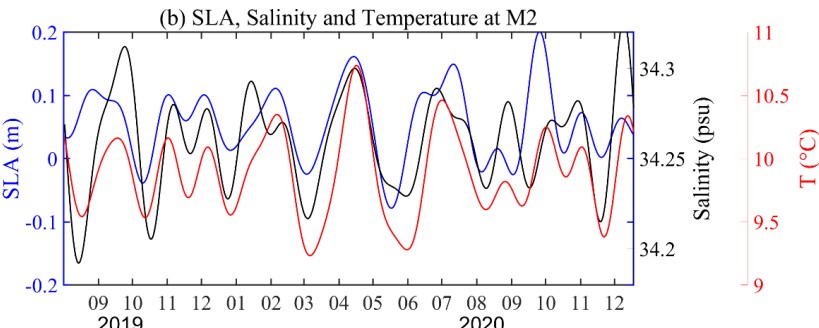


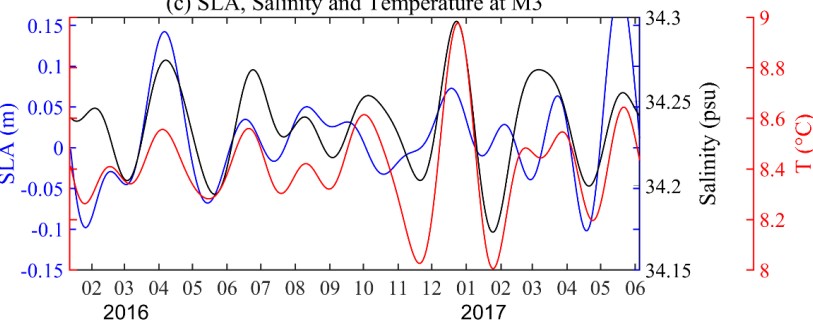


Figure 6. (a) The bandpass-filtered time series (20–120 days) of sea level anomalies
(SLAs; blue curve), vertically averaged temperature (red curve), and salinity between
500–800 m (green curve) at station M1. (b) and (c) same as (a), but for M2 and M3.


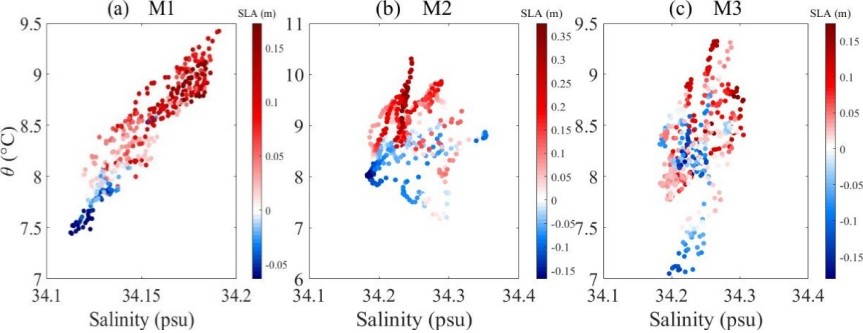






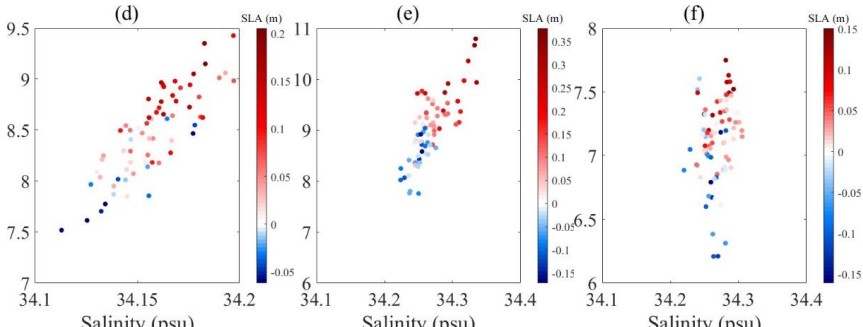

Figure 7. (a) The salinity and temperature data at M1 are displayed in a T–S scatter diagram with colors denoting the associated sea level anomalies (SLAs). (b) and (c) are same as (a), but for M2 and M3. (d), (e) and (f) are T-S plots of 500-800 m averaged temperature and salinity data from CMEMS product in the intermediate layer corresponding to the locations of M1, M2, and M3, respectively.

## 4 Disscussion

### 4.1 Structural Relationship Between Thickness and Salinity of NPIW

To further quantify the internal structure of the NPIW, linear regressions were performed between the thickness of the intermediate layer and the isopycnal-averaged salinity (between 26.4 and 26.9 σθ) at the three mooring sites. The derived regression equations are as show in Fig.8, where H represents the NPIW thickness (in meters) and salinity is the isopycnal-averaged salinity (in psu). The corresponding correlation coefficients between layer thickness and salinity are –0.63, –0.91, and –0.90 for M1, M2, and M3, respectively, indicating a strong and statistically significant inverse relationship at M2 and M3, and a weaker but still evident negative correlation at M1. All confidence intervals fall within the 95% confidence level, confirming the statistical robustness of these relationships. The relatively lower correlation at M1 may be attributed to multiple factors. The NPIW core at M1 is located at greater depths (typically deeper than 600 m), the salinity at M1 remained consistently below 34.2 psu throughout most of the observation period, exhibiting limited temporal variability. This reduced salinity fluctuation diminishes the sensitivity of the thickness–salinity relationship, thereby weakening the linear correlation. In contrast, M2 and M3 are located in regions characterized by more dynamic water mass interactions, including the influence of Kuroshio, South China Sea Intermediate Water (SCSIW), and saline subtropical waters from the western tropical Pacific. The resulting thermohaline variability enhances the responsiveness of NPIW structure to salinity changes, thereby strengthening the statistical coupling between thickness and salinity.

These regression relationships reinforce the potential of using isopycnal-averaged salinity as a structural proxy for intermediate water thickness, especially in regions or




datasets where direct thickness estimates are unavailable. This proxy relationship
provides a valuable way for reconstructing historical thickness changes or interpreting
reanalysis products in structural terms. More importantly, since mesoscale eddies
actively modulate both salinity and vertical structure in the intermediate layer, the
strong salinity–thickness coupling offers an indirect yet effective framework for linking
eddy-induced thermohaline variability to volumetric changes in the NPIW.
Although the water mass structure at M2 and M3 is more complex, the enhanced
salinity variability in these regions yields a more stable and robust relationship with
NPIW thickness. This suggests that the intermediate water structure is highly sensitive
to salinity perturbations and may respond coherently to mesoscale dynamical processes.

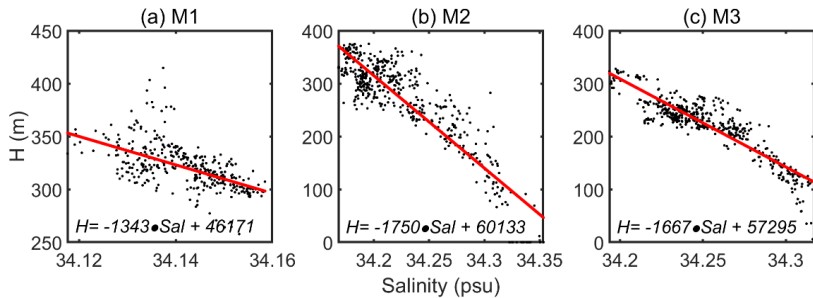


**Figure 8.** Scatterplots showing the relationship between NPIW layer thickness and
isopycnal-averaged salinity (26.4–26.9 σθ) at mooring sites M1, M2, and M3. Each
point represents a daily-averaged value. Linear regression fits are shown in red line.
The correlation coefficients are –0.63, –0.91, and –0.90 for M1, M2, and M3,
respectively.
**4.2 Observed Intraseasonal Variability of NPIW Induced by Mesoscale Eddies**
The observed presence of an ~80-day intraseasonal signal in sea level anomalies (SLA),
consistent across moorings M1 to M3, suggests that mesoscale eddies can exert
substantial influence on the NPIW properties. Wavelet analysis reveals that SLA,
salinity, and thickness variations all share similar periodicities in the range of 60–80
days, which aligns with the westward-propagating signal of mesoscale eddies in this
region.
Time series analysis further demonstrates that the salinity within the isopycnal layer
(26.4–26.9 σθ) exhibits clear intraseasonal oscillations, which are strongly anti-
correlated with variations in the thickness of the NPIW layer. This inverse relationship,
with correlation coefficients exceeding –0.9 at M2 and M3, indicates that salinity is not
only a tracer but also a reliable structural proxy for thickness variability in the
intermediate layer. The westward propagation of SLA bands during eddy events was
evident in longitude-time plots across all mooring latitudes (Fig. 9a, c, e), and lagged
correlations between SLA and salinity (Fig. 9b, d, f) confirmed the 60–80 day
propagation signals, with maximum correlation coefficients of 0.61, 0.5, and 0.6 at M1,
M2, and M3, respectively. The observed eddy signatures were further supported by case
analyses, in which anticyclonic (cyclonic) eddies were associated with increased



(decreased) salinity and NPIW thickness.

To further illustrate these patterns, we selected characteristic events exhibiting significant salinity and temperature changes. At mooring M1, for instance, two representative events were identified: a high-salinity episode on October 15, 2017 (Event 1), and a low-salinity episode on November 29, 2017 (Event 2). Satellite observations during these periods revealed the presence of an anticyclonic eddy during Event 1 and a cyclonic eddy during Event 2 (Fig. 10a, 10b). At both M2 and M3, similar associations between eddy polarity and salinity/thickness anomalies were observed (Fig. 10c–f), reinforcing the notion that eddy polarity (cyclonic vs. anticyclonic) plays a significant role in driving both hydrographic and structural variability in the NPIW.

These results indicate that mesoscale eddies are a dominant source of intraseasonal variability in the intermediate layer, influencing both the hydrographic properties (salinity) and vertical structure (thickness) of the NPIW. The coupled response highlights the necessity of considering both parameters when diagnosing water mass evolution under eddy forcing.

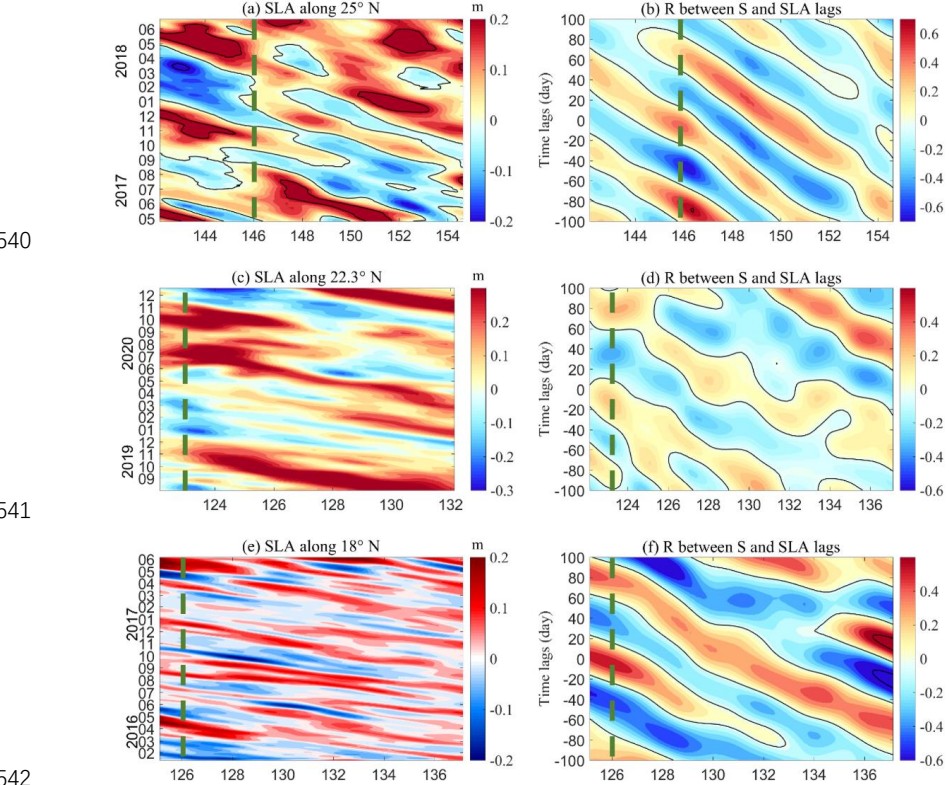

Figure 9. (a) Temporal evolution of sea level anomalies (SLAs) across longitudes at

25°N, as illustrated by the contour plots.; (b) The correlation coefficient between
salinity at M1 and SLA at different time lags, the vertical coordinates -100 to 100 days
in (b) represent SLA lagging salinity for 100 days and SLA exceeding salinity for 100
days, respectively. (c) and (e) are same as (a), but its along 22.3°N and 18°N,
respectively. (d) and (f) are same as (b), but for salinity from M2 and M3, respectively.
Black contours in Fig. 9b, 9d, and 9f represent the zero isolines. The green dash line
represent the location of M1, M2 and M3, respectively.




Figure 10. (a) and (b) are selected SLAs and surface geostrophic current maps
corresponding to the moments of Event 1 and Event 2 observed from M1, respectively,
where time of Event 1 corresponds to October 15, 2017 in Fig. 6a, and Event 2
corresponds to November 29, 2017 in Fig. 6a. (c) and (d) are same as (a) and (b), but
for mooring site M2, where time of Event 1 and Event 2 at M2 corresponding to April
20, 2020 and March 5, 2020 showed in Fig. 6b. (e) and (f) are same as (a) and (b), but
for mooring site M3, where time of Event 1 and Event 2 at M3 corresponding to April
10, 2016 and April 15, 2017 showed in Fig. 6c. The green dots denotes the mooring site,
the colors shading represent the SLAs and the arrows indicate the surface geostrophic
current.

**4.3 Mechanisms of Structural Modulation of NPIW by Mesoscale Eddies**
Although the variations in temperature and salinity at several moorings are correlated
with mesoscale eddies, it is challenging to understand from a broader perspective how
mesoscale eddies influence temperature and salinity changes in intermediate layer at
different regions. To further investigate the mechanisms by which mesoscale eddies
modulate the intraseasonal variability of NPIW, we employed CMEMS reanalysis data
as a complementary dataset to verify the robustness of our observational findings.



Power spectral analyses of 500–800 m averaged salinity at the mooring locations (Fig.
11a–c) revealed significant intraseasonal signals with dominant periods of 60–80 days,
consistent with those derived from mooring data. This confirms that the intraseasonal
variability of salinity in the intermediate layer is a robust signal and is well captured by
both in situ observations and reanalysis data. Furthermore, we compared scatter plots
of SLA against temperature and salinity derived from both mooring observations (Fig.
8a–c) and CMEMS reanalysis data (Fig. 8d–f). The CMEMS results exhibit similar
positive correlations between SLA and temperature/salinity, reinforcing the reliability
of the dataset for representing the hydrographic properties and eddy-induced variations
in NPIW. These consistencies justify the subsequent use of CMEMS data to support the
eddy mechanism analysis.

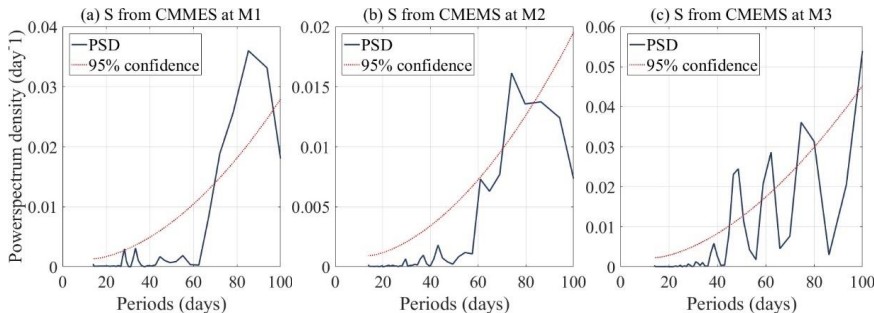

Figure 11. (a) The power spectrum density of 500-800 m averaged salinity at location
of M1 from CMEMS data. (b) and (c) are same as (a), but for location of M2 and M3,
respectively. The red dash line represent the 95% confidence level.

To further visualize the spatial structure, we calculated the horizontal distribution of
salinity and NPIW thickness using CMEMS data showed in Fig.12 to Fig. 14. The
thickness was defined as the vertical distance between the upper and lower boundaries
of the 34.3 psu isohaline, representing the volumetric extent of the low-salinity core.
Fig.12 to Fig.14 present the composite spatial distributions of intermediate-layer
salinity and NPIW thickness at the three mooring sites. These maps clearly demonstrate
that mesoscale eddies not only modify the thermohaline characteristics of the NPIW
but also significantly reshape its vertical structure. From a horizontal perspective,
anticyclonic eddies are generally associated with elevated salinity, whereas cyclonic
eddies correspond to lower-salinity regions. Meanwhile, the spatial distribution of layer
thickness differs notably among M1–M3, consistent with the mooring-derived
thickness variations, where M1 exhibits a generally thicker intermediate layer than the
other sites. Around anticyclonic eddies, the NPIW layer tends to become thinner, while
cyclonic eddies help preserve the low-salinity characteristics of the NPIW, resulting in
a thicker layer compared to anticyclonic conditions. This spatial correspondence aligns
with the strong inverse correlation between salinity and thickness identified in the time
series, further confirming that layer thickness also as an effective indicator of the eddy-
induced modulation of NPIW structure.





The modulation of NPIW by mesoscale eddies exhibits clear spatial heterogeneity
among the three sites. At M1, the intermediate water responds weakly to anticyclonic
eddies, showing smaller salinity variations. This weaker response is primarily attributed
to the deeper position of the NPIW core at this site, where anticyclonic eddies mainly
induce vertical displacement of isopycnals. Owing to the absence of surrounding high-
salinity water sources, the resulting property changes are limited. Nevertheless, the
observed downward displacement of isopycnals during anticyclonic periods still
indicates a vertical adjustment of the water mass, suggesting that mesoscale eddies
participate in the structural evolution of the NPIW in a more gradual and stable way. In
contrast, the M2 and M3 sites, located near the western boundary, exhibit much stronger
responses. These regions are influenced by complex water mass interactions involving
the SCSIW and Kuroshio Intermediate Water (KIW). Strong anticyclonic eddies in
these areas can substantially compress the intermediate layer, and through horizontal
advection and mixing entrain high-salinity waters from adjacent sources, leading to a
marked salinity increase and even temporary disappearance of the NPIW signature.
This indicates that mesoscale eddies in boundary regions not only alter the thermohaline
structure but can also reshape or redistribute the intermediate water itself.
These spatial features demonstrate that mesoscale eddies modulate NPIW through a
dual mechanism: vertical displacement governs thickness variation, while horizontal
advection and mixing amplify salinity anomalies—particularly in boundary mixing
zones. The combined effects of these processes generate the observed intraseasonal co-
variability of salinity and thickness. The inclusion of thickness analysis thus provides a
more comprehensive dynamical framework for understanding how mesoscale eddies
reshape NPIW properties, revealing their three-dimensional regulatory influence on
intermediate-water structure.

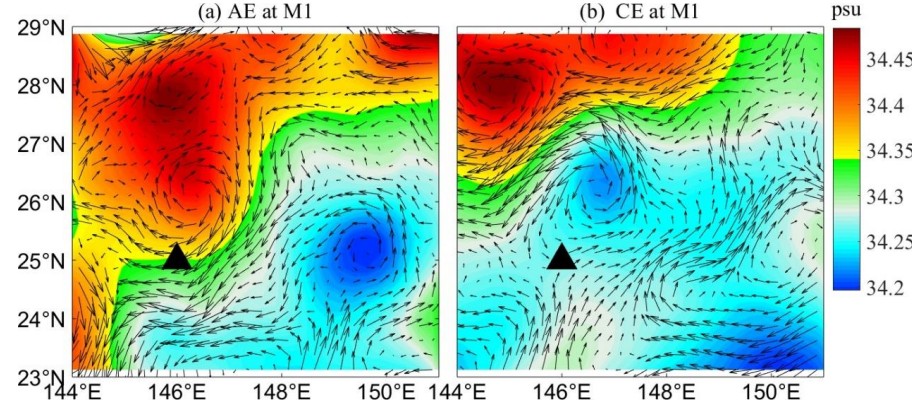




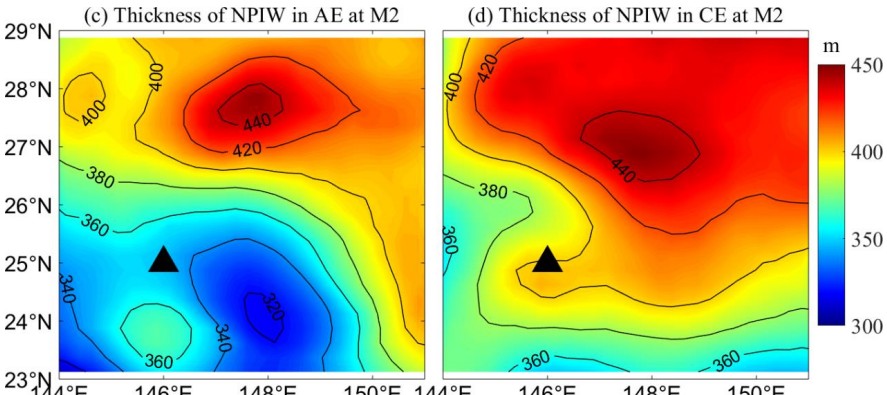

Figure 12. (a) Composite distribution of salinity (color shading) and current vectors (blue arrows) averaged over the 500–800 m layer during a high-salinity event near the M1 mooring site. (b) same as (a), but for low salinity event. (c) Thickness of NPIW at high salinity event, which calculated between 34.3 isohaline. (d) same as (c), but for low salinity event. The black triangle in figure represent the mooring location.

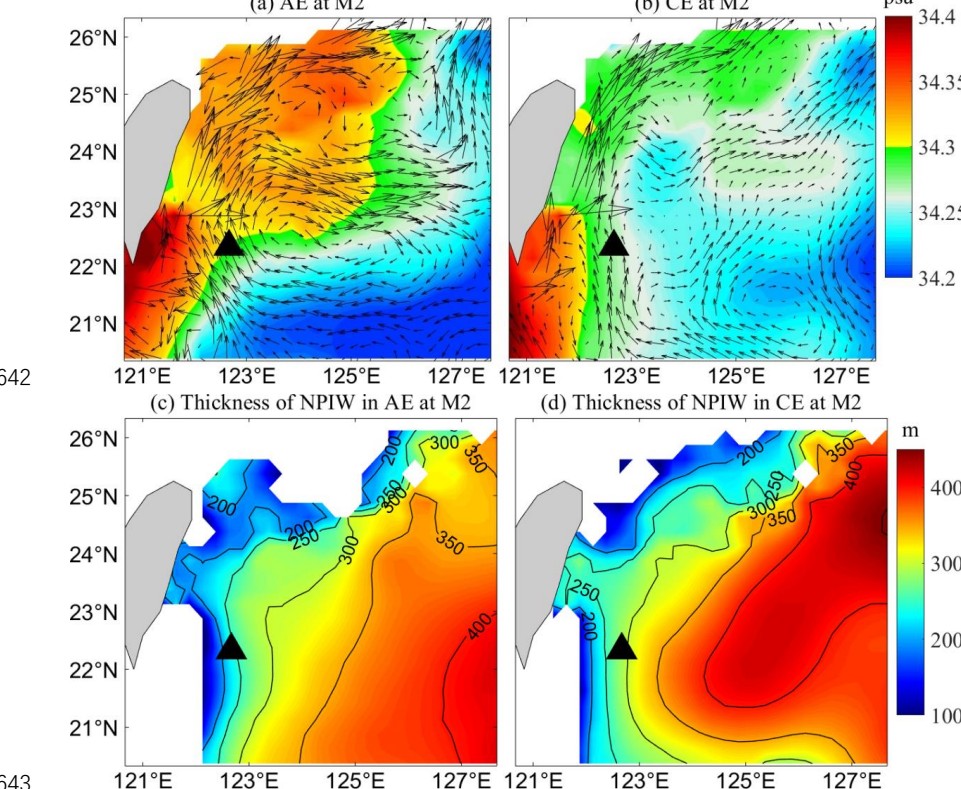

Figure 13. (a) Composite distribution of salinity (color shading) and current vectors (blue arrows) averaged over the 500–800 m layer during a high-salinity event near the





M2 mooring site. (b) same as (a), but for low salinity event. (c) Thickness of NPIW at
high salinity event, which calculated between 34.3 isohaline. (d) same as (c), but for
low salinity event. The black triangle in figure represent the mooring location.

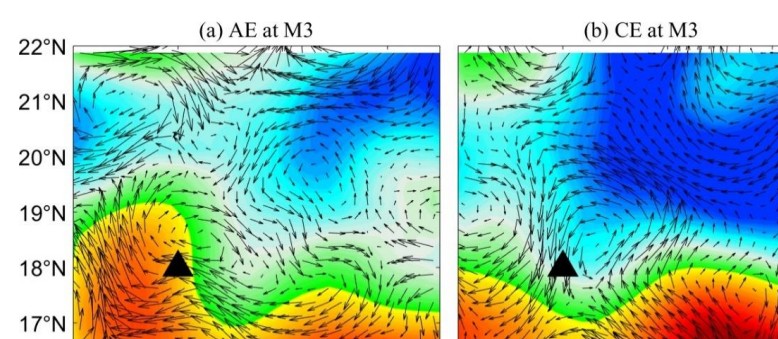


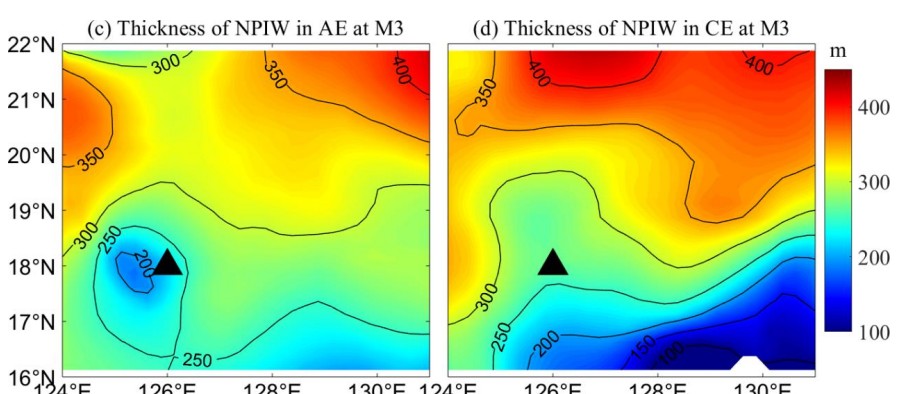


Figure 14. (a) Composite distribution of salinity (color shading) and current vectors
(blue arrows) averaged over the 500–700 m layer during a high-salinity event near the
M3 mooring site. (b) same as (a), but for low salinity event. (c) Thickness of NPIW at
high salinity event, which calculated between 34.3 isohaline. (d) same as (c), but for
low salinity event. The black triangle in figure represent the mooring location.

## 5 Conclusion

This study based on three long-term mooring observations (M1–M3) and reanalysis
data in the western Pacific, systematically investigates the intraseasonal variability of
the North Pacific Intermediate Water (NPIW) and its modulation by mesoscale eddies.
Wavelet analysis reveals a consistent ~80 days periodicity across all sites, and satellite
altimetry confirms that these intraseasonal signals are primarily induced by westward-
propagating mesoscale eddies.
A major innovation of this study is the introduction of NPIW layer thickness as a
structural diagnostic index, it quantitatively characterizes the compression and



expansion of the NPIW under eddy forcing. All three moorings show a strong inverse
correlation between layer thickness and salinity, indicating a tight coupling between
thermohaline anomalies and structural variations. This new metric provides a more
comprehensive framework for describing NPIW evolution beyond temperature–salinity
anomalies alone.
Composite maps further show that anticyclonic eddies correspond to higher salinity and
thinner intermediate layers, whereas cyclonic eddies produce lower salinity and thicker
layers. This spatial coherence highlights the role of eddy-induced vertical compression
and uplift in modulating NPIW structure. In the core of NPIW region, the eddy impact
is mainly vertical and moderate, while in the western boundary region, where multiple
water masses (SCSIW, KIW) interact, stronger eddies not only alter NPIW thickness
but also introduce high-salinity waters through horizontal advection and mixing,
occasionally leading to the disappearance of the NPIW signature.
In summary, by integrating multi-site observations with a new structural diagnostic
approach, this study reveals how mesoscale eddies control the structure and property
variability of NPIW. The inclusion of layer thickness provides a novel and physically
grounded perspective, extending beyond previous single-site analyses and enhancing
our understanding of intermediate-water dynamics in the western Pacific.

**Competing Interests Statement:** The authors have no conflicts of interest to declare.

**Acknowledgments**
This work is supported by the National Natural Science Foundation of China (No.
42206032), and the Natural Science Foundation of Shandong Province (No.
ZR2022QD045). We would like to thank all the personnel of the R/V Science for their
contribution to the data acquisition.

**Author contributions**
Qiang Ren conceived the study, led the research design, and drafted the initial version
of the manuscript. Yansong Liu, Feng Nan, Ran Wang, Xinyuan Diao, Jianfeng Wang,
and Xinchuang Liu contributed to the study design and were responsible for field data
collection and organization. Shumin Tu and Wei Huang provided technical support for
mooring instrumentation. Fei Yu and Zifei Chen carried out data analysis and
interpretation. All authors reviewed and approved the final version of the manuscript.
**Data Availability Statement**
The WOA data are provided by NOAA's National Oceanographic Data Center and
available from website: https://www.ncei.noaa.gov/products/world-ocean-atlas.
The merged gridded altimetry data can be downloaded from the website:
https://doi.org/10.48670/moi-00145; This study has been conducted using E.U.
Copernicus Marine Service Information https://doi.org/10.48670/moi-00052;



Researchers interested in accessing the mooring data may contact the corresponding
author, who will facilitate access through a formal data request procedure. If required
by the journal, the authors commit to coordinating with the data owner to deposit the
dataset in a publicly accessible repository and to provide a DOI upon acceptance.

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

Minimum Zones to Climate Change Based on Observations. *Geophysical Research Letters*, **49**(6):
e2022GL097724.