# Peer review of "Intraseasonal Variability and Eddy-Induced Structural Modulation of the North"

_EGUsphere, 2025_

## Referee Comment (RC1)

This is my second review of the manuscript "Intraseasonal Variability and eddy-induced structural modulation of the North Pacific Intermediate Water revealed by multi-mooring observations". The authors have made many changes since last version, so I took it as a new paper for my review. I see that great improvements have been made to make the paper stronger. However, some points still to be addressed before I can recommend the acceptance of this manuscript. Some points that have been partly brought up in my previous review may still need some clarification.

**Major:**

[1] The authors have made quite some improvements in the introduction as suggested in my previous review. I have some further comments to paragraph one: Can authors be more specific about the implications and importance of NPIW? These texts in the current version is high-level, and it would be helpful to include some details. The texts can be used on many objects other than NPIW.

[2] Line 101-105: It helps clarity to define NPIW. The authors did mention the depth, TS, and density of NPIW in line 82-84. However, after reading these sentences in line 101-105, I only have a vague idea about how NPIW is defined, because NPIW seems to be defined in different criteria across literature. If the definition is not a consensus, then explain it and introduce different kinds of definitions. For example, water mass within 34.3 isohalines like Figure 2 and section 3.2 is considered as NPIW. In Figure 3, the water mass with minimum temperature and salinity is NPIW. In section 3.2, the water mass between density of 26.4 and 26.9 is NPIW. It also helps to refer to a figure in the introduction about where NPIW is.

[3] The relationship between eddies, SLA, and NPIW thickness need some clarification. In line 352, if the inverse relationship between thickness and salinity is due to mesoscale eddies, does it mean that AE has fresher core and CE and saltier core given their isopycnal shape (see the schematic diagram below)? For example, is it right to consider that AE has convex lens-shaped isopycnals and thus increase the thickness, therefore the core salinity should be fresher than surroundings so that salinity decreases? Given that the NPIW has fresher cores, should the mixing of water mass always increase the salinity? Then can I interpret that the decrease in core salinity is not related to water mass mixing? The correlation analyses between TS and SLA seems to show that high S corresponds to high SLA, which may suggest that AE has saltier cores. How to reconcile these points?

[Figure]

**Minor points:**

Line 85: helps to refer to figure 1 to let readers know the location of this region

Line 103: I don't understand what this sentence means here, with the grammar mistake.

Line 111-113: This sentence seems to state that previous studies mostly focus on long time scales, and yet long-term observations are limited. Then what previous studies use to study the NPIW variability over long time scales? Models? This point needs to be clarified.

Line 184-188: can authors explain in more detail about the fixed-depth? Previous sentence suggests that CTDs are installed at 100 m vertical spacing, so the fixed depths are every 100 m?

Line 189: Do the adjacent CTD sensors refer to spatial domain? So the interpolation is done between the three morrings?

Figure 3 legend: the black line should be M1 not M2.

Figure 7 lower panels: add "theta" to the y axis of (d).

Line 472: It's interesting that in M1, correlation between salinity and layer thickness is weakest, but correlation between salinity and SLA is the strongest. One would assume that eddies have imprints on SLA, so that SLA and layer thickness should be comparable if eddies are considered. How to reconcile these two points? Maybe it helps to see depth-time figure with isopycnals and TS for individual eddy events.

**Grammar (not exhaustive):**

Line 82: remove "is"

Line 83: grammar problem

Line 93: should be period between "research" and "many", or change "," to ";". One rule to help is that only one verb should be found in one sentence, and each sentence should have one verb. Many such grammar mistakes are found throughout the manuscript.

Line 194: should be "deleting".